# Optimising Online Peer Support for People with Young Onset Dementia

**DOI:** 10.3390/ijerph21010060

**Published:** 2024-01-02

**Authors:** Esther Vera Gerritzen, Martin Orrell, Orii McDermott

**Affiliations:** Institute of Mental Health, Mental Health and Clinical Neuroscience, School of Medicine, University of Nottingham, Nottingham NG7 2TU, UK; martin.orrell@nottingham.ac.uk (M.O.); orii.mcdermott@nottingham.ac.uk (O.M.)

**Keywords:** Young Onset Dementia, online peer support, social health, interviews

## Abstract

People with Young Onset Dementia (YOD) can be hesitant to engage with online peer support. This work aims to explore (1) why people are hesitant to engage in online peer support, (2) how to get more people involved in online peer support, and (3) what makes online peer support work well. Nine interviews with people with YOD were conducted on MS Teams. Participants were recruited through purposive sampling. Data were analysed thematically. Reasons for being hesitant to engage with online peer support include being unsure what to expect and concerns about seeing others in more advanced stages of dementia. Additionally, it can be difficult to identify groups that suit one’s needs and interests. Group facilitators of online peer support groups should provide a detailed description of their group so that people can better assess whether the group would suit them. The insights obtained from this study will be used to develop a Best Practice Guidance on online peer support for people with YOD. Moreover, the findings can be useful for further research exploring how to support people with dementia in general in accessing online health and social care services.

## 1. Introduction

It is estimated that over 70,000 people in the United Kingdom (UK) are living with Young Onset Dementia (YOD) [1,2] (onset of dementia before the age of 65; van de Veen, et al. [3]). Bannon, et al. [4] and Stamou, et al. [5] found that people with YOD want and need services that support and respect their independence, identity, skills, and abilities, and that help them and their families in staying socially connected. Such an approach is in line with the Social Health Framework, which views health as a person’s ability to adapt and self-manage, and states that people can perceive good health and wellbeing despite having a chronic condition [6]. Dröes, et al. [7] found that when people with dementia incorporate positive coping strategies and focus on their abilities, they can still live a meaningful and fulfilling life. To maintain one’s social health, it is important that people with dementia have a strong support network that provides social support and inclusion, as well as care and support from informal and formal carers [7]. One way to improve social health and address the support needs of people with YOD can be through peer support. Peer support offers opportunities to stay socially connected and exchange experiential knowledge and information with others who are in a similar situation [8,9]. Stamou, et al. [10] and Cations, et al. [11] found that peer support can help people identify other relevant support services. Furthermore, through peer support, people can get involved in a variety of activities, such as those related to arts, crafts, or hobbies, but also research, policymaking, and advocacy. This allows people to keep using their skills and abilities [12,13]. Moreover, being involved in such activities can give people a sense of purpose again [8].

Nevertheless, many people with YOD do not have access to ongoing, age-appropriate, specialised YOD services [14], including peer support [15], as availability of such services is often inconsistent and short-term [16]. Online peer support could offer a solution, as it overcomes geographical barriers [17]. Research shows that there are many positives about online peer support. People can join from the comfort of their own home, it offers an opportunity to stay socially connected during the COVID-19 pandemic [12,18], and the wide variety of platforms that can be used for online peer support can tailor towards different needs and preferences [19]. However, our recent online survey showed that many people with YOD were hesitant to engage with online peer support [19]. Furthermore, symptoms such as speech and language difficulties or vision impairments, which are more common in younger people with a rare form of dementia, may make it more challenging for people to engage with technology or engage in online communication [20]. This work aims to (1) develop an in-depth understanding of why some people are hesitant to engage in online peer support, (2) explore what could be carried out to get more people involved in online peer support, and (3) develop an in-depth understanding of how online peer support could be optimised.

## 2. Materials and Methods

This qualitative study consisted of interviews with people with YOD. The interviews were conducted remotely during June–September 2022. The findings are reported following the consolidated criteria for reporting qualitative research [21]. This study received ethical approval from the London Bromley Research Ethics Committee (reference number: 21/LO/0248).

### 2.1. Recruitment and Eligibility Criteria

Participants were recruited through the online survey [19]. Out of 69 respondents, 56 people expressed interest in being involved in further parts of the study. The aim was to conduct 10–15 interviews or reach a point of data saturation, as it was not possible to conduct more interviews due to limited time and resources. Therefore, a sample of 19 people were invited. The sample was selected taking into account diversity and representativeness of the total sample. People from diverse groups (e.g., ethnicity other than White British, people living alone, or people in paid employment) were prioritised. The rest of the sample was made to create a balance of gender, age, time since diagnosis, living situation, and experience with online peer support.

### 2.2. Consent Procedures

For the interview, participants received a Participant Information Sheet and Informed Consent Form via email or post. Due to the wide variety of geographical locations of the participants (including one international participant), consent was taken remotely. The informed consent process was offered in different formats to accommodate individual needs and preferences. Participants could provide written or verbal consent. Written consent could be done by signing the consent form and sending it back to the first author, E.V.G. (via post or digitally). For verbal consent, E.V.G. went through the study information and consent form over a video call on MS Teams or a phone call, which would be recorded (after the participant gave permission to do so). Both options were presented to the participants, and all chose the option for written consent, sending it back digitally.

### 2.3. Interview Procedures

Participants were offered an interview via a videocall on MS Teams or a phone call. All participants chose for the videocall on MS Teams. All interviews were conducted by E.V.G. The interviews were semi-structured using a pre-defined interview guide (Table 1). The interview guide was informed by the findings from the survey [19]. The interview guide was sent to the participants in advance, so that they could prepare if they wanted to.

### 2.4. Data Collection and Analysis

The interviews were screen- and audio-recorded using the recording function and automatically transcribed verbatim using the transcribing function in MS Teams. E.V.G. checked the transcripts for accuracy and listened back to the recording to adjust where necessary. The recordings and transcripts were automatically saved on the password-secured University of Nottingham OneDrive account of E.V.G. The data were analysed through a thematic analysis using an inductive approach. The thematic analysis was conducted using the procedures outlined by Braun and Clarke [22] and consisted of six phases: (1) familiarising with the data, (2) coding the data, (3) developing initial themes, (4) developing and reviewing themes, (5) refining, defining, and naming the themes, and (6) writing up.

#### 2.4.1. Phase 1 and 2: Familiarizing with ad Coding the Data

During the first phase, E.V.G. read the transcripts and discussed initial thoughts and aspects that stood out with O.M., an experienced qualitative researcher. For the second phase, E.V.G. looked at the transcripts again in more detail and refined the initial thoughts and ideas, to then translate these into codes. During this phase, codes are specific and detailed parts of the transcript that are potentially interesting and relevant [22].

#### 2.4.2. Phase 3, 4, and 5: Developing and Reviewing Themes, and Writing Up

For the third phase, E.V.G. went back to the data and generated initial themes. Themes differ from codes in that they describe a broader meaning rather than focusing on specific elements from the transcript [22]. E.V.G. discussed the initial themes with O.M. and adjusted where necessary. For the fourth phase, E.V.G. applied the themes to all transcripts, to see if the themes captured the important elements and relationships within the data. After completing the application, E.V.G. discussed the process and refined the themes with O.M. and M.O. during phase five. Finally, during phase six, E.V.G. took the lead in writing up the findings, with detailed input from O.M. and M.O.

### 2.5. Trustworthiness of the Data

Triangulation can ensure trustworthiness of the data. Multiple methods of data collection were used: audio and screen recording, and field notes [23]. The audio matched the body language that was observed during the interview. The field notes highlighted the most important or striking elements that came forward during each interview, which was especially helpful in the first two phases of the analysis process. The research team consisted of researchers with different levels of experience and different professional backgrounds, which contributed to investigator triangulation [23].

## 3. Results

Of the nineteen people who were invited, five never responded to the invite and two expressed interest but did not respond again to schedule a date for the interview, and for three people, their supporter responded to let the research team know that the person with dementia was unable to take part in an interview due to the progression of their symptoms. This resulted in nine people taking part in an interview. After nine interviews, the research team discussed whether more interviews were necessary. The conclusion was that after nine interviews, participants did not introduce new topics anymore and data saturation was reached. Thus, the research team decided not to invite more people for an interview. An overview of the participant characteristics is presented in Table 2.

People described peer support as their lifeline and how it brought them hope and positivity after often a very difficult time after their diagnosis. On the other hand, people who did not use online peer support said that often they did not know how to find support that suits them, and some had experiences where the support did not meet their needs and interests. Five overarching themes and ten subthemes were generated (Table 3).

### 3.1. Theme 1: Looking for Support after the Diagnosis and Managing Life with YOD

Most participants received little to no information about peer support when they received their diagnosis. Many spent a long time after their diagnosis with no support at all, while trying to find some on their own. One person contrasted her experience with her mother’s, who also has dementia.

“My mum is in her 80s and she’s had someone coming to her house to do testing of words and memories and stuff like that. But I’ve not had any of that.”(female, 54, living with partner and other family members)

#### 3.1.1. Finding Support That Matches One’s Needs, Abilities, and Interests Can Be Difficult

For some who had non-Alzheimer’s dementia, it was difficult to find others with a similar diagnosis. Others struggled to find people of a similar age or with similar interests. People described their journey in finding the right support group for them, and some were still looking for it. One person who was still working said she was hoping to meet other people with YOD who were also still working.

“I’m testing the waters of different groups to see where I fit in. Unfortunately, I haven’t found one where I fit in and I’m comfortable, but I’m sure that there might be one someday, so it’s just keep on trying. Keep positive.”(female, 57, living with partner)

#### 3.1.2. Low Levels of Understanding of and Signposting to Peer Support Services by Healthcare Professionals

Many participants did not receive information from their doctor regarding peer support and had to search for this themselves. Some felt that their doctor did not have an understanding of the importance of peer support.

“I got some information from Dementia Mentors and Dementia Alliance International, and I had them send me some brochures, so that I could help distribute them where I am. I brought them to my doctors, but they had no interest. And I’m like: ‘this is basically saving my life’.”(female, 50, living alone)

#### 3.1.3. The Impact of Living with a YOD Diagnosis

For many, the diagnosis came as a shock and was followed by significant changes in their lives. Some heard how many years they would likely have left to live, and many had to give up their jobs. People also experienced a lack of understanding from family and friends.

“The one thing about this shocking news in your midlife is: you don’t know who to talk to. Some people might just think: ‘Oh, she’s got dementia, why doesn’t she talk to her mum?’. Well, my mum, she’s much more declined than I am, and she’s an old lady and she got Alzheimer’s. I think a lot of people don’t realise that Alzheimer’s is so different.”(female, 54, living with partner and other family members)

### 3.2. Theme 2: Barriers That May Stop People from Using Online Peer Support

One of the barriers to online peer support was difficulties with internet connection. One person who lived rurally said in-person support services can be far away, so that sometimes online is the only way to connect with others. However, poor-quality internet connection can be a problem.

“Online has the issue of connectivity, because we get three mega seconds and that’s good for up here. People get frustrated with the speed. People start talking over you but you’re still talking, due to the speed of the network.”(male, 63, living with partner)

#### 3.2.1. Online Peer Support Not Meeting Someone’s Needs, Abilities, or Interests

Participants in the earlier stages of dementia said they felt that services were tailored towards the needs of older adults. Some wanted specific information, for example, on continuing working. Others had tried online peer support, but it was not helpful.

“There was only two other people with dementia on the call and like five or six technical people, like doctors or nurses. We did that for probably about three months and then it just became a bit irrelevant because it was just going into a lot of technical detail about the whole thing, rather than just some sort of basic guidance as to what we should be doing.”(male, 60, living with partner and other family members)

Dementia symptoms can make it difficult to use technology for online peer support. For example, one participant needed support from his wife when attending video meetings. Another person said videoconferencing platforms were difficult and that they missed in-person, real human contact.

“I’m not meeting these people. It’s not like a real person. It’s like talking to a screen all the time. And so I stopped doing the groups because it was affecting me mentally; I think I just needed people contact rather than a screen.”(female, 57, living with partner)

#### 3.2.2. Feeling Unsure of What to Expect

Participants were hesitant to join because of being unsure of what to expect or feeling anxious about potentially seeing others in a more advanced stage of dementia. They said it was important to have similar interests and be of a similar age.

“I’m frightened of what I might see there [dementia cafes]. All the people that use those are further on than me, and I would perhaps feel like a fish out of water in a sense. That ‘so what am I doing here?’. ‘What can they do for me?’ type of thing. Because I don’t need something like that.”(female, 67, living with partner)

### 3.3. Theme 3: Navigating Challenges with Technology and Online Peer Support

People experienced challenges with using technology and online peer support and found alternatives or ways to cope. One person who had PPA (Primary Progressive Aphasia) and his wife explained that participating in group conversations, particularly online, was challenging due to difficulties with speech and recognising faces. They turned to YouTube videos as an alternative.

“They’re [people with or caring for someone with PPA] all putting their own experiences of things that have gone wrong and ways they’ve solved it, and activities. Some things we’ve picked up on and some things we thought ‘no, that’s not for us’. Even if we can’t attend the session in person, I’ve been able to follow a link later on if it’s being recorded, and then we can just tap into that information that way.”(male, 64, living with partner)

Another person said there are certain times where “her brain works better”, which is when she tries to engage in online peer support or research activities.

“There are times where I’ll forget how to turn my laptop on […]. Sometimes just nothing will make sense when I’m looking at the technology. And so, I just close the laptop and [think] ‘You know what, tomorrow’s gonna be better’.”(female, 50, living alone)

Online peer support can come with communication challenges, particularly for text-based communication on social media, for example, Facebook and Twitter. One person shared his political opinion on Twitter:

“I literally got hundreds of horrible tweets, every day for about a week. I am really careful now what I tweet because everybody can see your tweets. I don’t tweet much about my dementia, because again, you don’t know what response you’re going to get.”(male, 55, living with partner and other family members)

### 3.4. Theme 4: The Role of the Facilitator in Making Online Peer Support Work Well

A key aspect to make online peer support work well is having a good facilitator. People expressed that it is important that everyone has an opportunity to speak, that the facilitator should allow for the conversation to flow rather than speaking too much themselves, and that the facilitator should ensure that the meeting is a positive experience for everyone.

“Sometimes I’ve been on a group and it’s been one person talking an awful lot, and then I just think ‘what’s the point?’ and I don’t bother holding my card up. Or sometimes I may have forgotten what I wanted to say by the time it comes around for me to say something.”(female, 66, living with partner)

Participants said they believed part of the role of a facilitator was to support people in accessing the online meeting. This includes timely reminders, being available to provide technological support when needed, and providing instructions on how to join the meeting.

“I did a research project with [name of university] and we spoke to the facilitator beforehand. And she was wonderful. She asked us all what would help us in a peer support group, and she made a note of everybody individually about our needs and about the things that needed to be addressed for us.”(female, 66, living with partner)

Participants not involved in online peer support said it would be helpful to know what to expect from the group and whom it is for.

“It would be helpful to know what to expect when you go in rather than ‘oh well, we do our projects’. Ok well, so what does that mean? Just get a general thing about ‘this is a meeting place for people with dementia’. Do you have anybody with mild dementia? I don’t even know if there is anything out there for people like me.”(female, 67, living with partner)

### 3.5. Theme 5: Wider Opportunities for In-the-Moment Support

Most people appreciated exchanging social support, experiences, and information through online peer support. Participants felt a sense of mutual understanding and acceptance in their online peer support network, and some shared they have made new friendships.

“We talk about anything and everything. And we laugh together, we cry together, and most of all, it’s a safe place. It was something to look forward to every week and we became friends. You know, you can make friendships over videocalls as well as in person.” (female, 66, living with partner)

“In the group, you can just relax. Sometimes we can’t put our words together. Doesn’t matter, we all know what it’s like. Or times that we’re speaking almost normally, people celebrate that with you. Finding them was just one of the true highlights of my life.”(female, 50, living alone)

#### Advantages of Online Peer Support

People shared some of the advantages of online peer support compared to in-person peer support. Depending on the platform, support can be readily available at the moment when someone needs it. One person shared that his peer support group also has a WhatsApp group.

“Last week, I had a bit of a moment. At 4am, I wanted to go for a walk. I don’t know where, I don’t know why. I put that on my [group’s name] WhatsApp group and it was great, because I got five or six replies ‘don’t worry about it, it’s part of dementia’. It was nice to be able to put that on the WhatsApp group and somebody responds.”(male, 55, living with partner and other family members)

Furthermore, people can take part from the comfort of their own home. One person shared that not everyone in her environment knew about the diagnosis yet. With the meeting being on Zoom, she could turn off her camera and just listen in, and in that way stay anonymous. Others said they do not feel comfortable being in large groups, so that it was nice being able to join from their own home.

“We’re all like in the same room together, but without all that stimulation that you have when you’re in a room with nine other people. So Zoom, it’s just such a blessing.”(female, 50, living alone)

Online peer support can also overcome geographical barriers, as there is no need to travel.

“I think COVID has almost helped in that sense, because we couldn’t physically go to a meeting on London. We’ve done it through [video meeting] sessions and actually that’s been slightly more convenient. [Participant’s name] hasn’t been able to just hand in his notice and resign from his job, we’re still running a family business.”(male, 64, living with partner)

## 4. Discussion

This study provides new insights into why people with YOD may be hesitant to engage in online peer support. Through individual interviews, we could generate a deeper understanding of the underlying reasons for engaging or not engaging with online peer support, and what important skills for and responsibilities of online group facilitators are. These findings add more detail to our earlier survey findings [19].

### 4.1. Key Findings

This study highlighted that people with YOD experienced social support and friendship building through online peer support. Participants also emphasized unique benefits of online platforms, such as not having to travel, the opportunity for support right in the moment when they needed it, and not having all the stimulation that people may have when they are in a room full of people. These findings are in line with previous research on online peer support for people with chronic conditions [24], Parkinson’s disease [25], and Multiple Sclerosis [26].

However, this study also shows that people with YOD can face different barriers to accessing online peer support. For example, participants said that it is difficult to locate peer support that is appropriate to their needs and interests and that there is little support and signposting from healthcare professionals. Furthermore, some participants experienced a lack of consistency in support. Research shows that this is not only true for (online) peer support, but for support services in general [16]. An analysis of the Dutch post-diagnostic care and support system for YOD shows that consistency is one of the key elements in providing successful post-diagnostic support for people with YOD and their families. An example is through training and education for healthcare professionals and regional centres that are responsible for delivering YOD services, and which work together with local partners [27].

Another barrier to online peer support for people with YOD was not knowing what to expect from it. This is common in online peer support and has been identified in earlier research (for example, Multiple Sclerosis [28] and Polycystic Ovary Syndrome [29]). For example, some said they felt anxious about potentially seeing others who are in a more advanced stage or experience more severe symptoms and that they would like to know whether the group is attended by people in the early stages. Additionally, people may want to know what kind of activities or topics are usually on the agenda of the group. The current study also shows that there are some misconceptions among people with YOD about what (online) peer support entails, as some said that online peer support is only for older people or for those who are in a more advanced stage of dementia. We also identified this in our focus group study [12].

Furthermore, this study found that people with YOD can experience difficulties when using technology or when engaging in online communication and some may need support from others. This also came forward in a recent study on how people with a rare form of dementia and their carers were impacted during the COVID-19 pandemic. The study found that for many people it was difficult to engage with technology and have social gatherings and healthcare appointments online. It also showed that it could be very challenging for informal carers to find a way to support the person in engaging with technology. This can be due to the nature of symptoms of rarer forms of dementia, which may include difficulties in speech and language, and vision [20].

To address these barriers and make online peer support more accessible, participants said that it would be helpful to have more information about the group beforehand. A good group description should include information on the age range of the people who attend, whether it includes people who are newly diagnosed or not, whether people who attend are working (in paid employment, or in roles of volunteering, research, policy, or advocacy), and what people can expect from a meeting (e.g., the kind of topics that are discussed and the way a meeting normally goes). This is in line with one of the core principles that defines whether someone is a peer: sharing similarities [30,31,32]. The importance of similarity in peer support has also been highlighted in previous research. Lieberman, et al. [33] found that people with Parkinson’s disease who were in a homogenous support group (either based on time since diagnosis or age) felt more positive about their group compared to those in heterogeneous groups. Similarly, in their research on online peer support for informal carers of people with dementia, Han, et al. [34] found that the similarities shared made people feel understood and motivated to actively take part in the online group.

One of the key elements to make online peer support work well is having a skilled facilitator. Participants shared their views on what they think a good facilitator should do. For example, the facilitator should get to know people beforehand, and obtain an understanding of their needs and wishes. In this way, the person with dementia can find out if the group is something for them, and the facilitator can find out about any support the person with YOD may need. This was also found in an online group intervention for carers of people with a rare form of dementia [35]. Furthermore, the facilitator should ensure everyone has a chance to speak, and that it is a safe and confidential space. This is in line with previous research on the role of moderators in text-based online peer support communities. For example, research by Coulson and Shaw [36] shows that the moderators feel that it is their responsibility to create a sense of community and ensure that it is a safe space for everyone, welcome new members, and establish ground rules. Similarly, Huh, et al. [37] found that having a skilled moderator can help people feel safe in the online community, and that moderators can also help answer questions. However, some aspects of online facilitation may be different than facilitating in-person groups. For example, when online, it may be more difficult to read body language or clearly see facial expressions.

### 4.2. Limitations

This study only included people who were able to take part in a remote interview, through either MS Teams or a phone call. The reason for this was the wide geographical spread of participants, which made it not feasible to visit everyone in-person. Moreover, even if meeting in-person would have been possible in terms of distance and logistics, concerns around COVID-19 made this not preferential. Thus, the views and experiences of those who experienced significant barriers in using technology and engaging in online communication are not reflected in this study.

We aimed for 10–15 interviews and therefore invited a sample of 19 participants, assuming some would not be able to take part in an interview. However, just over one-third of the people who were invited to take part either never responded, or did not schedule a date for the interview, possibly because of the interviews taking place over the summer, during which people may not have had time to take part in research. This may have contributed to a smaller sample than we had aimed for. While we aimed for a diverse sample by giving people from less representative groups preference, the final sample was not as diverse as we had hoped (e.g., for ethnicity, employment status, or living situation). This reflects the population that we recruited from (people who took part in the online survey), which had a small proportion of non-White British participants, as well as a few people still in employment or living alone.

Furthermore, one must note that the interviews were conducted in the summer of 2022, towards the end of national lockdowns due to the COVID-19 pandemic. During the COVID-19 pandemic, online platforms such as Zoom became more widely used, which could have influenced how experienced people in this study were with online platforms. On the other hand, by the summer of 2022, many of the COVID-19 restrictions were lifted and people may have picked up their in-person activities again or gone on holiday, which could have influenced their interest in online peer support or taking part in online research.

### 4.3. Recommendations for Future Research

Future research could use the findings of this study to develop guidelines for facilitators on how to optimise online peer support for people with YOD and how to best support people online. Through qualitative methods such as surveys with open questions and interviews, we can explore whether peer support facilitators and moderators find the guidelines helpful and what improvements we can make. Additionally, such research could explore how others, for example, family members and friends, can identify support needs of people with YOD regarding using technology or remaining involved in online peer support groups. In this way, they can provide tailored support to the person with YOD without taking away their autonomy and independence. In line with this, future research could explore how people with different diagnoses of dementia experience online peer support. For example, this study included a person with PCA who described their difficulties in getting involved with online peer support due to the specific symptoms they experienced. People with rare, non-memory-led forms of dementia may have different support needs when using technology or engaging in online communication [20].

Scoping research can also explore whether implementation of the guidelines results in more online peer support groups being created, for example, by conducting a content analysis of dementia organisations or questionnaires among dementia organisations and NHS services for people with YOD. Moreover, as more and more health and social care services are being offered online, future research can go beyond YOD and peer support, and focus on how people with neurodegenerative conditions can be best supported in accessing online health and social care services.

## 5. Conclusions

Online peer support can be an important source of post-diagnostic support for people with YOD. To make it work well, it is key to have a trained and skilled facilitator, who listens, gives everyone a chance to speak, ensures the group is a safe space for everyone, and gets to know the members well. Additionally, this study recommends that facilitators of online peer support groups provide a detailed description of their group so that people can better assess whether the group would suit them. The insights obtained from this study will be used to develop a Best Practice Guidance on online peer support for people with YOD. Such guidance can raise awareness about what online peer support entails and what it could bring people, and can support organisations that are offering online peer support in optimising their services and provide better support to people with YOD.

## Figures and Tables

**Table 1 ijerph-21-00060-t001:** Interview guide.

General questions:
1. Do you use technology in your daily life? What do you use it for?
2. Did you ever experience difficulties when using technology? How did you deal with this?
3. Where do you go if you need support or information?
4. Have you used online peer support before? (For example, peer support in Zoom meetings, Facebook groups, Twitter, WhatsApp, or email)
If you have not used online peer support before:
5. Is this something you would consider using?
If you have used online peer support before:
6. What are the reasons for using online peer support?
7. What platforms do you use for online peer support? (For example, Zoom, Facebook, Twitter, WhatsApp, or email)
8. How does online peer support work for you?
9. What would you like to say to others who are considering getting involved in online peer support?
If you stopped using online peer support:
10. What are the reasons that you stopped using it?

**Table 2 ijerph-21-00060-t002:** Participant characteristics.

Male (%)	4 (44.4%)
Female (%)	5 (55.6%)
Age, mean (min–max)	59.6 (50–67)
Time since diagnosis	
<1 year (%)	3 (33.3%)
1–2 years (%)	3 (33.3%)
>3 years (%)	3 (33.3%)
Living situation	
Living with partner (%)	5 (55.6%)
Living with partner and other family members (e.g., children) (%)	3 (33.3%)
Living alone (%)	1 (11.1%)
Paid employment, yes (%)	2 (22.2%)
Ethnicity	
White—British (%)	8 (88.9%)
White—European (%)	1 (11.1%)
Experience with online peer support	
No—never used	3 (33.3%)
No—used before, but not currently	1 (11.1%)
Yes—current user	5 (55.6%)

**Table 3 ijerph-21-00060-t003:** Overarching themes and subthemes.

Overarching Theme	Subtheme
Looking for support after the diagnosis and managing life with YOD.	Finding appropriate support can be difficult. Low levels of understanding of and signposting to peer support services by healthcare professionals. The impact of living with a YOD diagnosis.
2. Barriers that may stop people from using online peer support.	Online peer support not meeting someone’s needs, and/or abilities, and/or interests. Being unsure of what to expect.
3. Navigating challenges with technology and online peer support.	Dementia symptoms impacting someone’s ability to use technology. Coping with challenges of online interpersonal communication. Need for technological support.
4. The role of the facilitator in making online peer support work well.	Organizational skills of the facilitator. Helping someone find the support that matches their needs and interests.
5. Wider opportunities for in-the-moment support.	

## Data Availability

For ethical reasons, the data collected for this study are stored securely at the University of Nottingham and are only accessible to the research team.

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
