# Peer review of "Optimising Online Peer Support for People with Young Onset Dementia"

_ijerph, 2024, doi:10.3390/ijerph21010060_

Round 1
Reviewer 1 Report
Comments and Suggestions for Authors
I believe this manuscript is a very interesting paper.
However, I would appreciate your could revise the following issues.
Line 164
Theme 5 in Table 3 is described as follows.
"Wider opportunities for in-the-moment support."
The result is as follows:
3.5. Theme 5: Integrating (online) peer support into someone's life
Why are these terms listed differently?
Also, Theme 5 has no subcategories.
Is this an incomplete result?
Line 360
The discussion section does not seem to have adequately addressed the five themes derived from this study.
Please consider this.
The following are minor issues.
EVG, MS, OM, and other abbreviations. What do these mean?
Please refer to the following and correct your references.
https://www.mdpi.com/authors/references
Also, please revise your references' expressions to previous studies in the Introduction section appropriately.
I hope you will find it useful.
Author Response
Reviewer’s comment |
Response |
1. Line 164 Theme 5 in Table 3 is described as follows: "Wider opportunities for in-the-moment support." The result is as follows: 3.5. Theme 5: Integrating (online) peer support into someone's life Why are these terms listed differently? Also, Theme 5 has no subcategories. Is this an incomplete result? |
Thank you for bringing this to our attention, these were different by mistake. We corrected this in the results to the following: “3.5. Theme 5: Wider opportunities for in-the-moment support”. The reason that there are no sub-categories for theme 5 is that during the data analysis process we did not find a need to include subthemes for theme 5, but instead all codes fitted well under the main theme. |
2. Line 360 The discussion section does not seem to have adequately addressed the five themes derived from this study. Please consider this. |
We added the following sections to address the themes that were not addressed yet in the discussion. Theme 5: (line 373-379) “This study highlighted that people with YOD experienced social support and friendship building through online peer support. Participants also emphasized unique benefits of online platforms, such as not having to travel, the opportunity for support right in the moment when they needed it, and not having all the stimulation that people may have when they are in a room full of people. These findings are in line with previous research on online peer support for people with chronic conditions [23], Parkinson’s disease [24] and Multiple Sclerosis [25].” Theme 1: (line 380-389) “However, this study also shows that people with YOD can face different barriers to accessing online peer support. For example, people with YOD said that it is difficult to locate peer support that is appropriate to their needs and interests and that there is little support and signposting from healthcare professionals. Furthermore, people experienced a lack of consistency in support. Research shows that this is not only true for (online) peer support, but for support services in general [16]. An analysis of the Dutch post-diagnostic care and support system for YOD shows that consistency is one of the key elements in providing successful post-diagnostic support for people with YOD and their families. For example, through training and education for healthcare professionals and regional centres that are responsible for delivering YOD services, and which work together with local partners [26].” Theme 3: (line 402-410) “Furthermore, this study found that people with YOD can experience difficulties when using technology or when engaging in online communication and some may need sup-port from others. This also came forward in a recent study on how people with a rare form of dementia and their carers were impacted during the COVID-19 pandemic. The study found that for many people it was difficult to engage with technology and have social gatherings and healthcare appointments online. It also showed that it could be very challenging for informal carers to find a way to support the person in engaging with technology. This can be due to the nature of symptoms of rarer forms of dementia, which may include difficulties in speech and language, and vision [29].” |
3. EVG, MS, OM, and other abbreviations. What do these mean? |
These are the authors’ initials, to indicate their contributions throughout the study. To clarify, we added the following to lines 83-84: “… the first author, EVG, …”. |
4. Please refer to the following and correct your references. https://www.mdpi.com/authors/references |
We downloaded the MDPI referencing style into our Endnote library and updated the references in the manuscript accordingly. |
5. Please revise your references' expressions to previous studies in the Introduction section appropriately.
|
We revised the references’ expressions to previous studies in the introduction and ensured they were accurate. |
Reviewer 2 Report
Comments and Suggestions for Authors
This is a qualitative study about a small group (n=9) of young-onset dementia patients who were asked about whether they would benefit from on-line peer support. There are several scientific short-comings with this study:
a) The sample size (n=9) is so small that it calls into question the generalizability of the findings. This needs to be addressed in the Discussion as a limitation.
b) In the Introduction, the authors neglect mentioning in para #1 the fact that dementia patients need the support of caregivers. They have, by definition, lost functional abilities, and they need caregivers to help them with hobbies and other activities. It is misleading to imply that an on-line support group would make their function improve. It is true that an on-line support group with other patients and caregivers would assist them to better understand what they are going through and would help them to feel less alone.
b) In the Results, the authors do not tell the reader about the diagnoses of their participants. This is a problem, since several YOD patients are likely to have difficulty with on-line encounters due to their unique symptoms (it would be good to mention these symptoms in para #2 of the Introduction, as well):
1) Posterior cortical atrophy patients have visuo-spatial problems.
2) Primary progressive aphasia patients have word-finding problems.
3) FTD patients have trouble with attention & concentration.
c) In the Discussion, the authors need to list the limitations of the study to include the small sample size, the lack of caregiver information, and the lack of dementia diagnoses. They also need to list the apparent paradoxes (the fact that 22% of participants were still receiving "paid employment"). Why would a "dementia" patient be employed?
Author Response
Reviewer’s comments |
Response |
1. The sample size (n=9) is so small that it calls into question the generalizability of the findings. This needs to be addressed in the Discussion as a limitation. |
In the original manuscript we addressed this in the limitations section (now lines 451-455). To emphasize it more we added the following on lines 455-456: “This may have contributed to a smaller sample than we had aimed for.” |
2. In the Introduction, the authors neglect mentioning in para #1 the fact that dementia patients need the support of caregivers. They have, by definition, lost functional abilities, and they need caregivers to help them with hobbies and other activities. It is misleading to imply that an on-line support group would make their function improve. It is true that an on-line support group with other patients and caregivers would assist them to better understand what they are going through and would help them to feel less alone. |
We added the following to the first paragraph in the introduction: (lines 33-35) “… it is important that people with dementia have a strong support network that provides social support and inclusion, as well as care and support from informal and formal carers [7].” |
3. In the Results, the authors do not tell the reader about the diagnoses of their participants. This is a problem, since several YOD patients are likely to have difficulty with on-line encounters due to their unique symptoms (it would be good to mention these symptoms in para #2 of the Introduction, as well): 1) Posterior cortical atrophy patients have visuo-spatial problems. 2) Primary progressive aphasia patients have word-finding problems. 3) FTD patients have trouble with attention & concentration. |
We did not collect this information from the people taking part in the interviews. The aim of this study was to get an understanding of participants’ experiences with online peer support in a broad sense, regardless of their diagnosis. Considering the wide range of diagnoses typically present among people with YOD and the typically small sample size of interview studies, it would have been difficult to draw conclusions based on specific subtypes of dementia. We acknowledge that this is an important point and therefore we addressed it in the discussion in 4.1 Key findings: (402-410) “this study found that people with YOD can experience difficulties when using technology or when engaging in online communication and some may need support from others. This also came forward in a recent study on how people with a rare form of dementia and their carers were impacted during the COVID-19 pandemic. The study found that for many people it was difficult to engage with technology and have social gatherings and healthcare appointments online. It also showed that it could be very challenging for in-formal carers to find a way to support the person in engaging with technology. This can be due to the nature of symptoms of rarer forms of dementia, which may include difficulties in speech and language, and vision [20]” and in 4.3 Recommendations for future research: (476-485) “Additionally, such research could explore how others, for example family members and friends, can identify support needs of people with YOD regarding using technology or remaining involved in online peer support groups. In this way they can provide tailored support to the person with YOD without taking away their autonomy and independence. In line with this, future research could explore how people with different diagnosis of dementia experience online peer support. For example, this study included a person with PCA who described their difficulties in getting involved with online peer support due to the specific symptoms they experienced. People with rare, non-memory led forms of dementia may have different support needs when using technology or engaging in online communication [20]” We added the following to the introduction: (lines 52-55) “Furthermore, symptoms such as speech and language difficulties or vision impairments, which are more common in younger people with a rare form of dementia, may make it more challenging for people to engage with technology or engage in online communication [20].” |
4. In the Discussion, the authors need to list the limitations of the study to include the small sample size, the lack of caregiver information, and the lack of dementia diagnoses. They also need to list the apparent paradoxes (the fact that 22% of participants were still receiving "paid employment"). Why would a "dementia" patient be employed? |
Please see our response to comment 1 regarding the small sample size. Regarding the caregiver information, we did not include this because the study focussed on the experiences of the person with dementia and not on the experiences of caregivers. However, we do acknowledge the important role of informal carers in supporting the person with dementia, and therefore we added the following to section 4.3. Recommendations for future research: (lines 477-481) “Additionally, such research could explore how others, for example family members and friends, can identify support needs of people with YOD regarding using technology or remaining involved in online peer support groups. In this way they can provide tailored support to the person with YOD without taking away their autonomy and independence.” Regarding the dementia diagnoses we added the following to section 4.3. Recommendations for future research: (lines 481-486) “future research could explore how people with different diagnosis of dementia experience online peer support. For example, this study included a person with PCA who described their difficulties in getting involved with online peer support due to the specific symptoms they experienced. People with rare, non-memory led forms of dementia may have different support needs when using technology or engaging in online communication [20]”. People with YOD are of working age, and thus it can happen that people with a YOD diagnosis are still in paid employment. Therefore, the authors support people with dementia who are still in employment. Employers and employees with YOD can work together to make reasonable adjustments to the workplace and tasks, so that someone may be able to stay in employment, despite their dementia diagnosis. |
Round 2
Reviewer 1 Report
Comments and Suggestions for Authors
I believe that this manuscript has been revised in an appropriate manner. Thank you very much.
Reviewer 2 Report
Comments and Suggestions for Authors
The authors have addressed the concerns I raised in my review. Thank you.